# Dendritic NMDA spikes are necessary for timing-dependent associative LTP in CA3 pyramidal cells

Federico Brandalise[1,2], Stefano Carta[1,2], Fritjof Helmchen[1,2], John Lisman[3] & Urs Gerber[1,2]

The computational repertoire of neurons is enhanced by regenerative electrical signals initiated in dendrites. These events, referred to as dendritic spikes, can act as cell-intrinsic amplifiers of synaptic input. Among these signals, dendritic NMDA spikes are of interest in light of their correlation with synaptic LTP induction. Because it is not possible to block NMDA spikes pharmacologically while maintaining NMDA receptors available to initiate synaptic plasticity, it remains unclear whether NMDA spikes alone can trigger LTP. Here we use dendritic recordings and calcium imaging to analyse the role of NMDA spikes in associative LTP in CA3 pyramidal cells. We show that NMDA spikes produce regenerative branch-specific calcium transients. Decreasing the probability of NMDA spikes reduces LTP, whereas increasing their probability enhances LTP. NMDA spikes and LTP occur without back-propagating action potentials. However, action potentials can facilitate LTP induction by promoting NMDA spikes. Thus, NMDA spikes are necessary and sufficient to produce the critical postsynaptic depolarization required for associative LTP in CA3 pyramidal cells.

[1] Brain Research Institute, University of Zurich, CH-8057 Zurich, Switzerland. [2] Neuroscience Center Zurich, University of Zurich, ETH Zurich, CH-8057 Zurich, Switzerland. [3] Department of Biology and Volen Center for Complex Systems, Brandeis University, Waltham, Massachusetts 02453, USA. Correspondence and requests for materials should be addressed to U.G. (email: gerber@hifo.uzh.ch).

The Hebbian postulate, whereby a synapse is strengthened when presynaptic input is successful in evoking post-synaptic activity, is accepted as the basis for many forms of associative learning. This process is initiated by strong depolarization of the postsynaptic neuron, which activates NMDA (N-methyl-D-aspartate) receptors and results in the calcium elevation that triggers the biochemical processes leading to long-term potentiation (LTP). The depolarization produced by a single synaptic input is insufficient to induce LTP; rather, LTP is associative in nature, requiring the summated input from many excitatory synapses. A key aspect of Hebbian plasticity is the requirement of a feedback signal that informs the synapse whether the integrated postsynaptic activity was sufficient to induce an action potential. An attractive candidate for this feedback event is the back-propagating action potential (bAP)[1,2]. However, a number of observations have called into question the relevance of the bAP for LTP[3,4]. Importantly, LTP can be induced at various synapses in the absence of bAPs[5–13]. In these studies, regenerative dendritic events referred to as dendritic spikes (dSpikes) may produce the depolarization needed to trigger LTP. These dSpikes can be local events, often confined to a single dendritic branch[14]. Theory suggests that such spatially restricted depolarization may have the advantage of allowing memory storage at a much finer scale than the widespread dendritic regions affected by bAPs[15]. At present, however, the evidence that dSpikes provide the critical depolarization necessary for LTP remains correlative.

Dendritic spiking mechanisms are complex, involving both bAPs as well as dSpikes (dendritic sodium spikes, dendritic calcium spikes and dendritic NMDA spikes[14]). Although all these events may influence dendritic depolarization, determining which factor is essential for LTP is complicated. We have investigated LTP at recurrent collateral synapses in the CA3 hippocampal region and characterized the underlying roles of bAPs and dSpikes. Using a combination of electrophysiological and two-photon $Ca^{2+}$ imaging techniques, we identify dendritic NMDA spikes as the causal signal that initiates LTP at synapses between hippocampal CA3 pyramidal cells.

## Results

### NMDA spikes induce branch-restricted $Ca^{2+}$ transients and LTP.
We evoked NMDA spikes, which are a class of dSpike dependent on NMDA receptor (NMDAR) activation[16,17], using a subthreshold input-timing-dependent plasticity (ITDP) protocol. CA3 recurrent (rCA3) axons were stimulated followed 10 ms later by mossy fibre (MF) stimulation (Fig. 1a). In these experiments, action potentials were prevented by intracellular application of QX-314 (500 μM)[18]. Sixty pairings at 0.1 Hz produced LTP at the stimulated rCA3 synapses ($41.9 \pm 2.3\%$ after 30 min, $n = 13$ of 16, $P < 0.001$; Fig. 1a), whereas MF EPSPs remained unaffected (Fig. 1b and ref. 9). Scaling membrane potential traces to normalize the amplitude of the evoked rCA3 EPSP preceding MF stimulation revealed a bimodal distribution of summated excitatory postsynaptic potentials (EPSPs) corresponding to linear and supralinear responses (Fig. 1c). The supralinear responses exhibited all-or-none properties as a function of stimulation intensity and were blocked by NMDAR antagonists, thus meeting two important criteria for NMDA spikes[19] (Supplementary Fig. 1 and Fig. 1d; see 'Methods' section for NMDA spike analysis). To facilitate analysis, all experiments were performed with intracellular picrotoxin (1 mM) to suppress GABA$_A$-mediated inhibition. However, ITDP-triggered LTP can also be induced in the absence of picrotoxin[9].

We next examined a third criterion for NMDA spikes, the localized elevation of intracellular calcium concentration.

Two-photon $Ca^{2+}$ imaging with Fluo-5F showed that during the pairing protocol $Ca^{2+}$ transients occurred in the region of the dendritic rCA3 synapses close to the stimulation electrode (Fig. 1e and Supplementary Fig. 2d), consistent with the finding that NMDA spikes are triggered by NMDAR expressed at rCA3 rather than at MF synapses[9]. Importantly, whenever an evoked dendritic $Ca^{2+}$ transient was detected, a coincident NMDA spike was likely to be present in the electrophysiological recording (in $93.8 \pm 3.5\%$ of cases, $n = 13$; Fig. 1f and Supplementary Fig. 2e). In addition, the integral of all $Ca^{2+}$ transients detected within the field of view close to the stimulation electrode correlated with the amplitude of the NMDA spike (Supplementary Fig. 3e–g). Dendritic $Ca^{2+}$ imaging also provided information about the localization and spatial extent of NMDA spikes. $Ca^{2+}$ transients were restricted to individual branches, spreading only for short distances (width: $11.8 \pm 1.2 \mu m$, $n = 13$ cells; Fig. 1e,h and Supplementary Fig. 2h–j). Furthermore, within a branch, there were local hotspots (∼1 μm diameter) that were repeatedly elicited during the pairing protocol, likely reflecting the sites of synaptic input ($n = 7/7$; Fig. 1h,i and Supplementary Fig. 2h–j), as reported previously[10,11,19–21]. Branch-specific $Ca^{2+}$ transients could be observed in apical ($n = 7$) or basal dendrites ($n = 9$) depending on the position of the stimulation electrode (see 'Methods' section for further description). This pattern reflects the location of rCA3 synapses, which can be on either apical or basal dendrites. The location of giant MF synapses is closer to the cell body, but still able to depolarize more distal dendrites[22].

Dendritic $Ca^{2+}$ transients can arise not only in conjunction with NMDA spikes but also with back-propagating APs, dendritic sodium spikes and $Ca^{2+}$ spikes. Under our experimental conditions, however, the dendritic $Ca^{2+}$ transients corresponded to NMDA spikes, on the basis of their duration (∼50 ms)[23,24], absence of a sodium spikelet[23,24] and restricted spatial propagation[19,24] (Supplementary Fig. 4). Furthermore, dendritic NMDA spikes were evoked with a lower stimulation threshold than $Ca^{2+}$ spikes, and were not prevented by pharmacological blockade of sodium and $Ca^{2+}$ channels[9,25,26]. Both the NMDA spikes and the corresponding dendritic $Ca^{2+}$ transients were greatly decreased in number after NMDAR blockade with the competitive antagonist D-AP5 (50 μM) (NMDA spikes: from $44.2 \pm 2.7\%$ in control, $n = 13$, to $4.8 \pm 1.7\%$, $n = 6$, $P < 0.001$; $Ca^{2+}$ transients: from $41.9 \pm 2.3\%$, $n = 13$, to $5.1 \pm 2.7\%$, $n = 6$, $P < 0.001$; Fig. 1d,g and Supplementary Fig. 2f,g). Taken together, the results demonstrate that these supralinear events are chiefly NMDA spikes localized to individual dendritic branches.

Are the NMDA spikes evoked by the ITDP protocol expressed in only one region of the dendritic tree or can multiple areas be implicated? We addressed this question by examining the EPSP/EPSC rise time as a proxy for the distance of the stimulated rCA3 inputs from the soma (Supplementary Fig. 5). In the majority of cells (10/13), the relatively low-intensity stimulation employed (minimal stimulation (60% failures) +20% in all experiments) evoked rCA3 EPSPs with unimodal distributions of rise time, which is consistent with the generation of the EPSP at a specific dendritic distance from the soma, even though a CA3 pyramidal cell typically receives multiple synaptic contacts from a given neighboring cell[27]. Plotting the rise time of the rCA3 EPSP against the mean distance from the soma of the observed $Ca^{2+}$ transients yielded a linear relation (Supplementary Fig. 5e). However, in 3/13 experiments, the $Ca^{2+}$ transients were detected in two distinct areas of the dendritic tree; consistently in these cases, the rise time analysis revealed a bimodal distribution (Supplementary Fig. 6).

A strong indication that NMDA spikes are important for the induction of LTP was obtained by examining the relationship between the probability of evoking dendritic $Ca^{2+}$ transients

characteristic of NMDA spikes and the magnitude of LTP. This revealed a high correlation ($r = 0.79$, $n = 16$; Fig. 1j). In experiments where paired stimulation evoked $Ca^{2+}$ transients in less than 10% of the 60 trials, rCA3 synapses did not undergo potentiation ($1.1 \pm 0.1\%$, $n = 3$, $P = 0.5$; Fig. 1j and Supplementary Fig. 7). Thus, the paired stimulation paradigm *per se* does not induce potentiation, but rather a critical number ($\sim 10$) of NMDA spikes is required. Furthermore, the number of branches exhibiting a dendritic $Ca^{2+}$ transient during a paired

stimulation correlated with the amplitude of the NMDA spike (Fig. 1k).

**NMDA spikes provide the postsynaptic depolarization for ITDP.**
We next sought to address the causal role of NMDA spikes in LTP induction by bidirectionally manipulating the probability of evoking NMDA spikes. In a first set of experiments, we examined whether LTP induction is facilitated by enhancing the

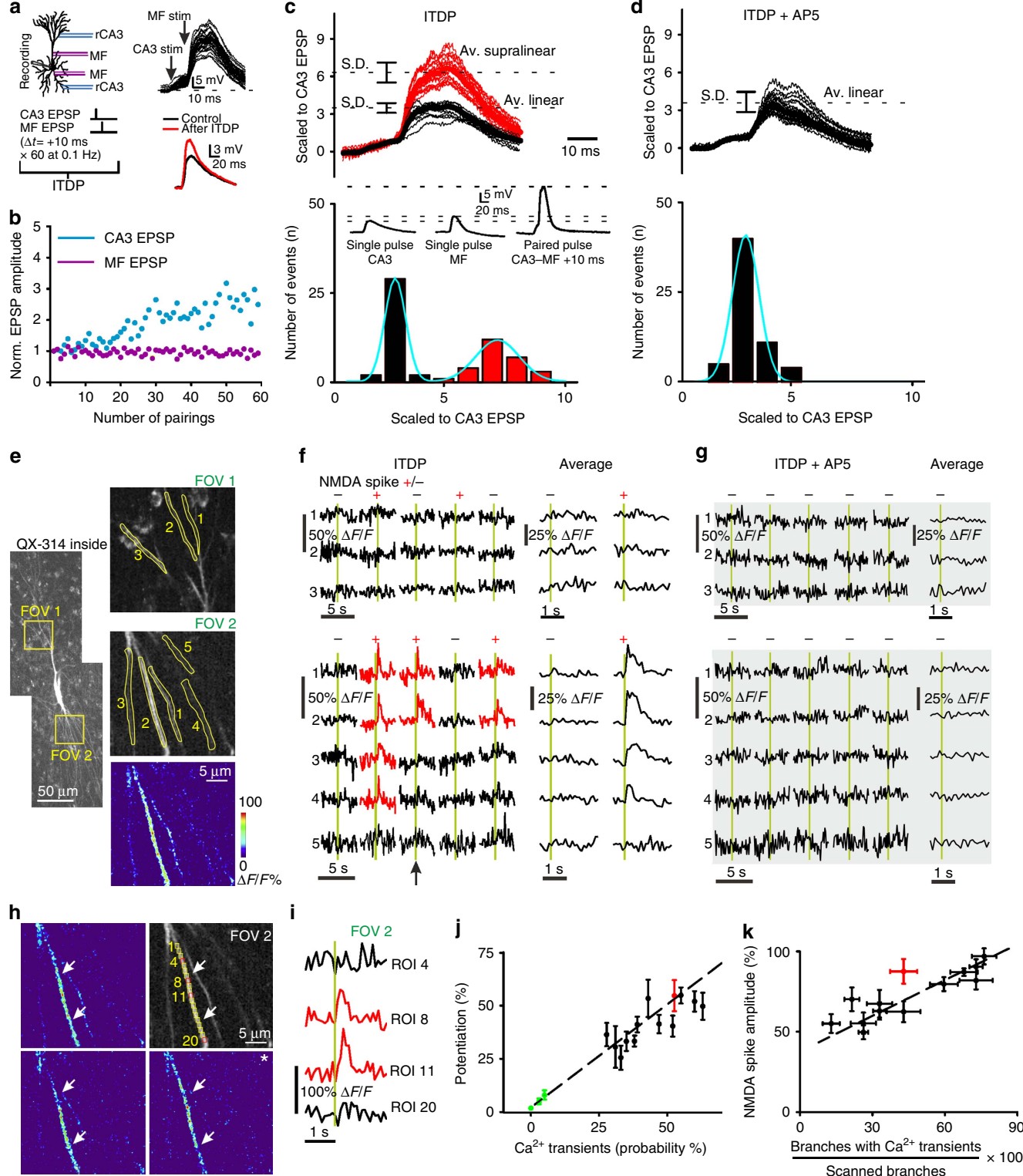

activation of extrasynaptic NMDARs, as these play a key role in the generation of NMDA spikes[24,25]. Using the above-described ITDP pairing protocol with sodium channels blocked by intracellular QX-314, we reduced the stimulation intensity below the threshold to evoke NMDA spikes reliably (in only $5.4 \pm 2.0\%$ of pairings, $n = 8$). Repetitive pairing for 60 times with this weak ITDP protocol failed to induce LTP ($0.9 \pm 0.5\%$, $n = 8$, $P = 0.6$; Fig. 2a,b). However, after inhibiting glutamate transporters with *threo*-β-benzyloxyaspartic acid (TBOA), a procedure that promotes glutamate spillover and activation of extrasynaptic NMDARs[28], stimulation with the same intensity significantly increased the probability of evoking NMDA spikes (in $46.4 \pm 2.6\%$ of pairings, $n = 8$, $P < 0.001$) and their normalized amplitude (from $95.3 \pm 13\%$ to $132.9 \pm 10.4\%$, $n = 8$, $P < 0.01$; Fig. 2b,c). Moreover, LTP was now observed at the rCA3 synapses ($65.1 \pm 4.3\%$, $n = 8$, $P < 0.001$; Fig. 2a,b).

In the next experiments, we reduced the probability of evoking NMDA spikes by hyperpolarizing the membrane potential and looked at the resulting change in the ability to induce LTP. At $\sim -60$ mV ($-60.5 \pm 3.4$ mV, $n = 7$), the pairing of rCA3 and MF inputs consistently resulted in NMDA spikes ($45.6 \pm 2.9\%$ of pairings, $n = 7$, $P < 0.001$), whereas at more hyperpolarized membrane potentials ($-74.2 \pm 2.3$ mV, $n = 7$), NMDA spikes were rarely evoked (in $6.1 \pm 1.5\%$ of pairings, $n = 7$; Fig. 2d–h). LTP could not be induced in the experiments with hyperpolarized membrane potential in which NMDA spikes were infrequent (rCA3 EPSP amplitude change: $-1.2 \pm 0.5\%$, $n = 7$, $P = 0.7$), while in the same cells at $-60$ mV, LTP was observed in 7/7 experiments ($58.1 \pm 2.9\%$, $n = 7$, $P < 0.001$; Fig. 2g).

**NMDA spikes provide the postsynaptic depolarization for STDP.** A dendritic event implicated in synaptic plasticity is the bAP[1,2]. However, it is possible that a bAP by itself has little direct effect on LTP induction and only becomes important if it triggers other dendritic events[3]. Because all of the above experiments were performed under conditions where action potentials were prevented, it was of interest to examine whether bAPs play a role in triggering NMDA spikes and if so, whether this process is reliable. We therefore paired the rCA3 EPSP with a back-propagating AP instead of a MF EPSP, which corresponds to a classical spike-timing-dependent plasticity (STDP) protocol. Simultaneous recordings were obtained from a second-order apical dendrite and the soma (Supplementary Fig. 8). The stimulating electrode was positioned so as to activate rCA3 fibres that targeted mainly the dendrite recorded from, as indicated by the faster rise time and the greater amplitude of the rCA3 EPSP

recorded in the dendrite versus the soma (dendrite: $6.1 \pm 0.5$ ms, soma: $9.2 \pm 0.8$ ms, $P = 0.008$; dendrite: $8.2 \pm 1.9$ mV, soma: $4.9 \pm 0.7$ mV, $n = 5$, $P = 0.007$; Fig. 3c). When this EPSP was followed after 10 ms by a single AP triggered with somatic current injection, an NMDA spike was never observed ($n = 5$; Fig. 3a,d). The contribution of NMDARs to the responses generated by this single-AP STDP protocol as compared with the ITDP protocol was small but not negligible ($13 \pm 1.6\%$ versus $102.1 \pm 7.9\%$, $n = 5$, $P = 0.0001$; Fig. 3d). Weak activation of NMDARs as occurs in response to such a single bAP might therefore not be sufficient to produce LTP, as reported in other cell types[29–32]. To test this possibility directly, we paired a single bAP with synaptic input over 50 trials at 0.3 Hz. As show in Fig. 3a,e, this protocol did not induce LTP ($2.2 \pm 2.4\%$, $n = 5$, $P = 0.7$). Thus, simply activating NMDARs may be insufficient for LTP induction if the generated response is below threshold to evoke an NMDA spike.

Previous studies in other cell types have shown that whereas STDP is difficult to evoke using a single bAP, brief bursts of bAPs can be effective[2,31,32]. When the rCA3 EPSP was paired after a 10 ms delay with three APs at 200 Hz, an NMDA spike of dendritic origin was evoked with high probability ($61.2 \pm 9.3\%$, $n = 5$; Fig. 3a,d). Repetitive pairing with this STDP protocol induced strong LTP ($44.9 \pm 9.9\%$, $n = 5$, $P = 0.0033$; Fig. 3a,e). Thus, AP firing has to attain a critical frequency to evoke NMDA spikes and to induce LTP (Supplementary Fig. 9).

To further clarify the role of bAPs in modulating the probability of evoking an NMDA spike, we tested a STDP protocol in which a hyperpolarizing pulse was applied through the dendritic recording electrode during the triggering of the three somatic APs, which prevented NMDA spikes (Fig. 3f,h). Importantly, the injected current did not hyperpolarize the membrane below the threshold for AP generation (Fig. 3f) and did not markedly reduce the 'non-regenerative' NMDA component of the response ($11.7 \pm 2.6\%$; $n = 5$, $P = 0.002$; Fig. 3h). Again, rCA3 fibres were stimulated that targeted mainly the dendrite recorded from, as indicated by the faster rise time (RT) and the greater amplitude of the rCA3 EPSP recorded in the dendrite versus the soma (dendrite: $5.6 \pm 0.6$ ms, soma: $8.7 \pm 0.7$ ms recording, $n = 5$, $P = 0.0008$). With this protocol, LTP was not induced ($3.76 \pm 3.84\%$, $n = 5$, $P = 0.3$; Fig. 3f,i), showing that the occurrence of bAPs *per se* is not sufficient to initiate synaptic plasticity. In the same cell, the omission of the hyperpolarizing pulse recovered NMDA spikes ($59.5 \pm 7.2\%$, $n = 5$; Fig. 3h) and repetitive pairing now induced LTP ($43.9 \pm 2.9\%$, $n = 5$, $P < 0.0001$; Fig. 3f,i). Taken together, these

**Figure 1 | Branch-specific NMDAR-dependent dendritic Ca$^{2+}$ transients evoked by subthreshold synaptic pairing are associated with LTP induction.** (**a**) Left: ITDP protocol for pairing rCA3 and MF inputs to a hippocampal CA3 pyramidal cell in slice culture. The recording pipette contained 500 μM QX-314. Right: representative example of pairing-evoked EPSPs and averaged traces of the rCA3 EPSP before and after LTP. (**b**) Time course throughout the pairing protocol of EPSP amplitudes normalized to baseline. rCA3-evoked but not MF-evoked EPSPs are potentiated. (**c**) Representative example of scaled voltage traces (normalized to initial rCA3 EPSP amplitude) reveal a bimodal distribution of response amplitude corresponding to linear (black) and supralinear (red) summation. Inset: individual traces for a rCA3 EPSP, a MF EPSP and a summated supralinear EPSP. (**d**) Supralinear responses are suppressed by NMDAR blockade (D-AP5) resulting in a unimodal distribution of summated EPSP amplitudes. (**e**) Fluo-5F labelled CA3 pyramidal neuron. Fluorescence measurements to detect pairing-induced Ca$^{2+}$ transients were obtained in three ROIs for apical dendritic branches in FOV 1 and five ROIs for basal dendritic branches in FOV 2. Lower right image shows localized Fluo-5F $\Delta F/F$ fluorescence change for one pairing trial (arrow in **f**). (**f**) Example Ca$^{2+}$ transients from ROIs selected in **e**, recorded during 5/30 representative consecutive pairings (green bars). Trials with linear ('−') and supralinear ('+') EPSP summation are indicated. Ca$^{2+}$ transients for FOV 2 were averaged separately for linear ($n = 14/30$) and supralinear ($n = 16/30$) trials. (**g**) NMDAR blockade abolished dendritic Ca$^{2+}$ transients as shown for the same ROIs as in **f**. (**h**) A series of uniformly sized ROIs ($\sim 1 \times 1$ μm) numbered from 1 to 20 were positioned along a responsive dendritic segment (delineated by arrows) as identified from the heat map in **e**. Images from three pairing trials in which an NMDA spike was evoked. (**i**) Ca$^{2+}$ transients associated with an NMDA spike for ROI 8 and 11 (red traces) and for ROIs outside the active region (black traces) for the image in **i** marked with an asterisk. (**j**) The magnitude of LTP of the rCA3 EPSP correlates across cells with the incidence of Ca$^{2+}$ transients during pairing (number of trials with Ca$^{2+}$ transients in at least one ROI divided by the total number of ITDP pairings). In cells where the pairing protocol failed to evoke Ca$^{2+}$ transients (green data points), EPSPs were not potentiated. Red data point corresponds to the example cell shown in **d** ($r = 0.79$, $n = 16$). (**k**) Pooled data showing the increase in NMDA spike amplitude as a function of the prevalence of dendritic Ca$^{2+}$ transients ($r = 0.73$, $n = 16$).

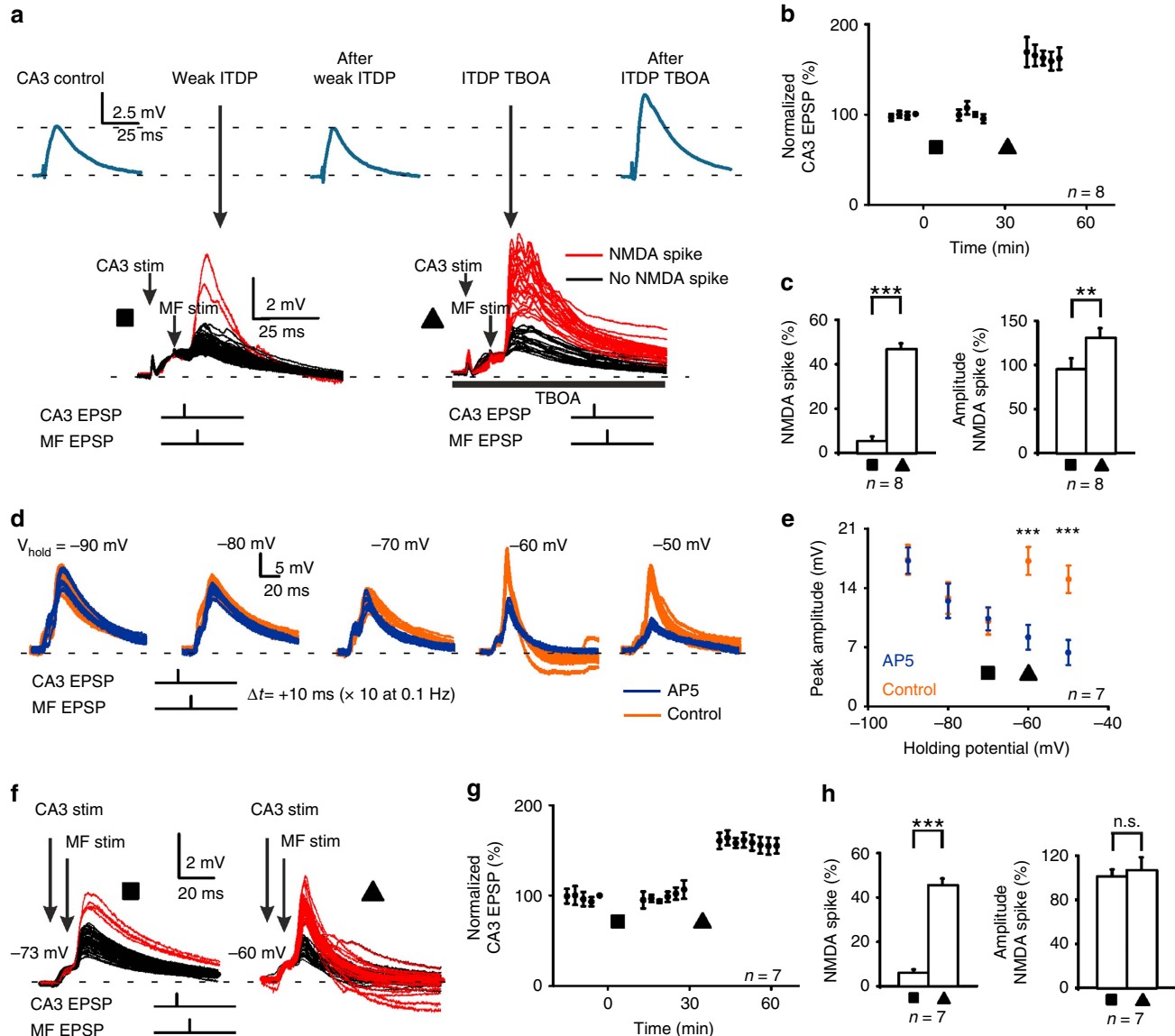

**Figure 2 | LTP induction can be manipulated bidirectionally by altering the probability of NMDA spikes.** (**a**) Representative example of voltage recordings during a weak ITDP protocol, which was below threshold for consistently evoking NMDA spikes and did not potentiate the rCA3 EPSP (top, blue traces). Increasing glutamate spillover with $10\,\mu M$ TBOA during a second pairing protocol in the same cell enhanced the probability of NMDA spikes (red traces) and induced LTP (square: ITDP; triangle: ITDP $+$ TBOA). (**b**) Pooled data for rCA3 EPSP amplitude after the weak ITDP protocol was applied in the absence and presence of TBOA (rCA3 EPSP amplitude was measured after TBOA washout). (**c**) Pooled data for the probability of evoking an NMDA spike and for the NMDA spike amplitude in the two conditions. (**d**) Decreasing NMDA spike probability by hyperpolarizing the membrane potential. An evoked rCA3 EPSP was paired with a subsequent MF EPSP at decrementing holding potentials with and without NMDAR blockade (D-AP5). For each condition, pairing was repeated only 10 times to avoid inducing LTP. (**e**) Pooled data show a linear decrease in summated EPSP amplitude during NMDAR blockade when the holding potential becomes more depolarized (blue traces). Repeating the experiment without NMDAR blockade reveals a nonlinear enhancement in EPSP amplitude at membrane potentials less negative than $-70\,mV$ (orange traces). (**f**) When the ITDP protocol was delivered at a hyperpolarized holding potential (more negative than $-70\,mV$, square), where NMDARs do not contribute significantly to synaptic responses, NMDA spikes were rarely evoked. When the same protocol was repeated at $-60\,mV$ (triangle), NMDA spikes were evoked with high probability. (**g**) Time course of rCA3 EPSP amplitude showing that the ITDP protocol has no effect at $-73\,mV$ (square) but induces LTP at $-60\,mV$ (triangle). (**h**) Pooled data for NMDA spike incidence ($6.1\pm1.5\%$ versus $45.6\pm2.9\%$, $n=7$, $P<0.001$) and amplitude ($101.3\pm4.9$ versus $106.9\pm11.8$, $n=7$, $P=0.68$, paired $t$-test) for the two conditions.

data show that NMDA spikes are the critical event in both subthreshold and suprathreshold forms of LTP at rCA3 synapses.

The bAPs associated with STDP protocols also activate voltage-gated $Ca^{2+}$ channels that may trigger dendritic $Ca^{2+}$ spikes[2,33,34]. After pharmacological block of $Ca^{2+}$ currents, however, the supralinear response persisted (Supplementary Fig. 9f,g).

## Discussion

We have presented several lines of evidence for a direct causal role of dendritic NMDA spikes in the induction of LTP at excitatory recurrent synapses onto CA3 pyramidal cells. We were able to evoke dendritic electrical signals definitively identified as NMDA spikes and to show that manipulations that increased the probability of NMDA spikes increased LTP, whereas

manipulations that decreased their probability reduced LTP. NMDA spikes were the critical trigger both when LTP was induced with subthreshold events using an ITDP protocol as well as with bAPs using a classical STDP paradigm. Thus, our results also provide new insights into the function of bAPs. According to

the text-book description of STDP, such spikes, when timed after the onset of EPSPs, provide the crucial depolarization to initiate LTP. However, for CA3 pyramidal cells, this timing rule is not absolute, depending on the age of the animal[35,36]. As shown here and previously[9], at synapses between CA3 pyramidal cells, LTP

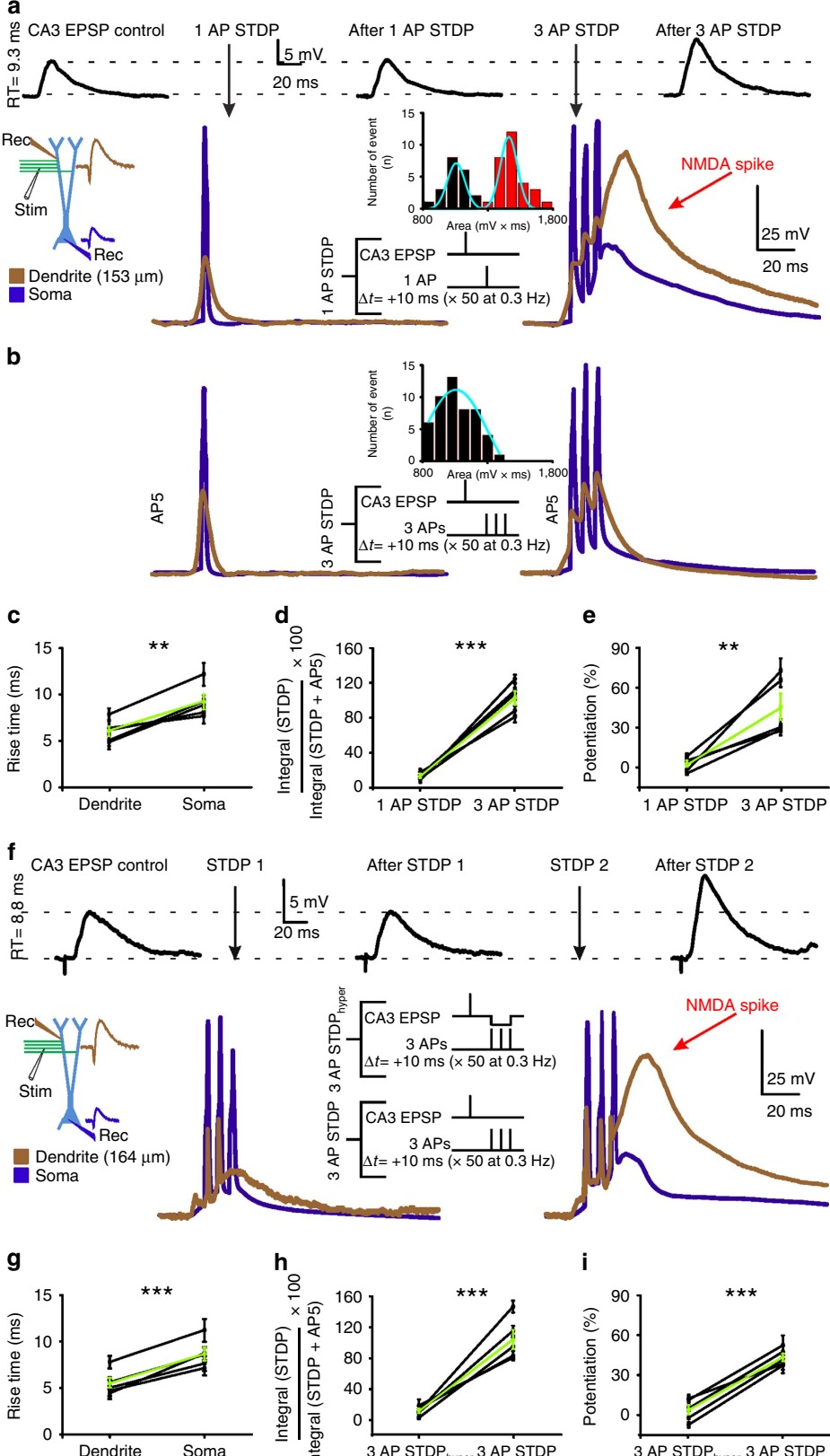

can be induced in the absence of bAPs. Nevertheless, bAPs, which can be boosted during concurrent synaptic input[37], may have an indirect role in inducing synaptic plasticity. In our study, brief bursts of bAPs evoked NMDA spikes, which in turn induced LTP. Thus, the NMDA spike rather than the bAP is the final effector of LTP at the CA3 synapses we have studied. Determining the postsynaptic depolarization necessary for LTP in rCA3 synapses is of particular significance because both theoretical and experimental work point to the critical role of this brain region in autoassociative memory[38–40].

Our findings raise the question whether NMDA spike-mediated subthreshold LTP induction represents a special property of recurrent synapses in the CA3 area or whether NMDA spikes may also be involved at other synapses. Previous work in diverse brain areas has shown that dendritic spikes generate the supralinear signal triggering plasticity at synapses where LTP can occur independently of bAPs[5,6,11–13]. Specifically for NMDAR-dependent dendritic spikes, in vivo investigations have provided correlative evidence for their role in synaptic plasticity in CA1 pyramidal cells[41], in motor cortex[10] and in barrel cortex[11]. Although these studies did not rigorously establish the unique role of NMDA spikes as in our experiments, they are consistent with our conclusion that dSpikes rather than bAPs provide the critical depolarization for LTP. Thus, NMDA spikes appear to be important for LTP at synapses in diverse brain regions, but other synapses have been shown to depend on other types of supralinear signalling[8,12–14]. It should be noted that in our preparation a single bAP almost never triggered an NMDA spike and did not induce LTP. However, it is possible that single spikes may be able to more powerfully affect NMDA spikes or other dendritic processes in vivo[42], where background neuronal activity and neuromodulatory conditions are different[43].

Our data indicate that on the order of 10 NMDA spikes (∼20% of the responses during the 60-trial protocol lasting 6 min) are sufficient to induce LTP, which is in the same range as the number of bursts that a hippocampal pyramidal cell fires as an animal traverses the cell's place field[44]. Significant LTP may thus be induced by a single experience. In this respect, it is of interest that the long-term representation of space by CA1 hippocampal place cells in navigating mice depends directly on the prevalence of dendritic branch spikes[41,45]. Our data provide further support for the concept of the dendritic branch rather than the dendritic spine as the functional unit for LTP

induction[46–48]. NMDA spikes in CA3 pyramidal cells are generated when ∼15 synapses are activated synchronously on a dendritic segment[20,49], which is a relatively small number considering that 15 synapses represents <5% of the synapses present in a dendritic segment[20]. This observation is consistent with the compartmentalization of correlated inputs onto single dendritic segments during hippocampal development[50–52]. Such clustered input may be a prerequisite for the generation of spatially restricted NMDA spikes and their critical role in LTP induction.

## Methods

**Preparation of acute hippocampal slices.** Acute slices were prepared from 3-week-old Wistar rats following a protocol approved by the Veterinary Department of the Canton of Zurich. (approval ID 81–2014). Rats were decapitated and brains quickly removed in an ice-cold artificial cerebrospinal fluid (ACSF) solution containing the following (in mM): 125 NaCl, 2.5 KCl, 1.25 NaH$_2$PO$_4$, 25 NaCHCO$_3$, 1 MgCl$_2$, 2 CaCl$_2$, 10 glucose (pH 7.4) and equilibrated with 95% O$_2$ and 5% CO$_2$. Three hundred micrometre thick transverse acute slices were prepared with a vibratome (HM 650 V, Microm International) in ice-cold artificial cerebrospinal fluid. Sections were incubated in ASCF for 20 min at 34 °C and then kept at room temperature for at least 1 h before recording. Experiments with acute slices are depicted in Supplementary Fig. 10.

**Preparation of hippocampal slice cultures.** All other experiments were performed in slice cultures, which form a quasi-monolayer that facilitates dendritic imaging. Slice cultures were prepared from 6-day-old Wistar rats according to the Gähwiler method[53]. Transverse slices were prepared (400 μm) and fixed to coverslips with clotted chicken plasma. These were placed in sealed test tubes with serum-containing medium and maintained in a moving incubator at 36 °C for 21–28 days.

**Patch-clamp recording.** Hippocampal slice cultures or acute slices were mounted in a recording chamber positioned on an upright microscope (Zeiss Axioskop FS1) or, for combined imaging, a Scientifica microscope. Slices or slice cultures were superfused with an external solution (pH 7.4) containing (in mM): 137 NaCl, 2.7 KCl, 11.6 NaHCO$_3$, 0.4 NaH$_2$PO$_4$, 2 CaCl$_2$, 2 MgCl$_2$, 5.6 D-glucose and 0.001% phenol red to monitor pH for slice cultures and 125 NaCl, 2.5 KCl, 1.25 NaH$_2$PO$_4$, 25 NaCHCO$_3$, 2 CaCl$_2$, 2 MgCl$_2$, 10 glucose (pH 7.4) and equilibrated with 95% O$_2$ and 5% CO$_2$ for acute slices. All experiments were performed at 34 °C. Patch pipettes had a resistance between 5 and 7 MΩ for somatic whole-cell recordings of CA3 pyramidal cells and between 9 and 11 MΩ for recordings from second-order dendrites. Both somatic and dendritic patch pipettes were filled with (in mM): 135 K-gluconate, 5 KCl, 10 Hepes, 5 phosphocreatine, 2 MgATP, 0.4 NaGTP and 0.07 CaCl$_2$ (pH 7.2). In experiments in which imaging was performed, Fluo-5F (100 μM) and Alexa 495 (10 μM) were added to the solution. Resting membrane potential of the hippocampal CA3 pyramidal cells was − 64.1 ± 2.9 mV, n = 53. Voltage commands were corrected for the liquid junction potential (8.3 mV). No differences were apparent in the properties of NMDA spikes recorded in slice cultures and in acute slices (Supplementary Fig. 10).

**Figure 3 | STDP protocols induce LTP only if NMDA spikes are generated.** (**a**) Representative voltage traces recorded simultaneously at the soma and a second-order dendrite (see schematic), first during an STDP protocol pairing rCA3 EPSPs with a single AP evoked by brief somatic current injection (2 ms; 4 nA; left), followed by pairing with three evoked APs delivered at a frequency of 200 Hz (right). A 1 AP STDP protocol failed to potentiate EPSPs, whereas 3 AP STDP resulted in LTP (top traces). Note that the 1 AP STDP protocol was generally insufficient to generate a supralinear dendritic response, whereas the 3 AP STDP protocol caused dendritic spikes visible in the dendritic recording and to a lesser extent, because of cable filtering, in the somatic recording. Inset: Area under the evoked responses was plotted to distinguish between linear and supralinear events. (**b**) Supralinear responses with the 3 AP STDP protocol were prevented following NMDAR blockade (D-AP5, 1.4 ± 3.2%, n = 5, P < 0.0001). Inset: supralinear responses are prevented when NMDARs are blocked. (**c**) Faster EPSP rise times in dendrite versus soma (n = 5, P = 0.008, paired t-test) indicates that for these experiments a majority of the synapses activated by stimulation of rCA3 collaterals were located at or near the dendritic branch recorded from. The green line denotes the pooled average of all cells. (**d**) The NMDAR contribution generated by the different STDP protocols was estimated by calculating the ratio between the areas under the voltage traces in the absence and presence of D-AP5 (n = 5, P = 0.0001, paired t-test). (**e**) When the STDP protocol did not adequately activate NMDARs to generate a supralinear response, LTP was not induced (n = 5, P = 0.7; paired t-test). (**f**) A hyperpolarizing pulse applied during the 3 AP STDP protocol decreased the probability of evoking an NMDA spike and prevented LTP induction. As a result, a supralinear response was not generated, either in the soma (blue trace) or the dendrite (brown trace). Furthermore, the rCA3 EPSP was not potentiated (black traces: rCA3 EPSP recorded in the soma). When the same protocol was repeated, but without the hyperpolarizing pulse, supralinear responses were generated and LTP was induced. (**g**) Pooled data showing faster rise times for the rCA3 EPSP in the dendritic as compared with the somatic recording (n = 5, P = 0.0008; paired t-test), indicating that primarily dendritic inputs were stimulated. (**h**) Quantification of the contribution mediated by NMDARs to the recorded responses shows that the STDP protocol that included a hyperpolarizing pulse (3AP STDPhyper) inhibited the generation of a supralinear signal versus without the hyperpolarization (3AP STDP, n = 5, P = 0.0001, paired t-test). (**i**) In addition, the STDP protocol that included a hyperpolarizing pulse greatly reduced the magnitude of LTP at the rCA3 synapse (n = 5, P = 0.0002, paired t-test).

For the determination of current–voltage relationships, command potentials had a duration of 1 s to ensure steady-state responses. Data were recorded with Axopatch 200B amplifiers (Molecular Devices), digitized at 4 kHz for current-clamp and 5 kHz for voltage-clamp, and analysed off line with pCLAMP 10 (MolecularDevices) and Origin (OriginLab). In all experiments, inhibitory postsynaptic potentials were reduced in the recorded CA3 pyramidal cells by adding picrotoxin (1 mM) (Sigma-Aldrich) to the patch solution. Series resistance (typically 8.5–14.5 MΩ for somatic recordings and 15–35 MΩ for dendritic recordings) was monitored regularly, and cells were excluded if a change of > 20% occurred during the recording. For experiments involving extracellular stimulation, the electrode placement, the paired pulse ratio, the response latency and the sensitivity of transmission to DCG-IV were assessed to differentiate between responses mediated by MFs versus rCA3 fibres (Supplementary Fig. 11).

**Stimulation paradigm.** CA3 pyramidal cells receive excitatory input in a spatially segregated manner:

(1) MFs, the axons of the dentate granule cells, contact the emerging trunk of the apical and basal dendritic tree[22].
(2) rCA3s, the axons of neighbouring CA3 pyramidal cells, target the intermediate region of the apical and basal dendritic tree[54].
(3) Perforant path axons, originating in the entorhinal cortex, contact the distal portion of the apical dendritic tree[54].

Consistent with this innervation pattern, our experiments show that $Ca^{2+}$ transients associated with NMDA spikes occur both in the apical and basal dendritic tree. Dendritic $Ca^{2+}$ transients were evoked mainly in the apical dendritic tree when the rCA3 stimulating electrode was placed in the stratum radiatum of CA3 and the MF electrode in the dentate gyrus. Dendritic $Ca^{2+}$ transients were evoked mainly in the basal dendritic tree when the rCA3 stimulating electrode was placed in the stratum oriens of CA3 and the MF electrode in the dentate gyrus. The identity of the stimulated fibre tracts was ascertained using standard criteria (Supplementary Fig. 11).

**NMDA spike analysis.** Supralinear responses, which correspond to NMDA spikes, were revealed by normalizing the amplitude of the evoked rCA3 EPSP preceding MF stimulation (for example, Fig. 1c). All 60 paired responses were scaled to the amplitude of the rCA3 EPSP revealing a bimodal distribution.

The probability of evoking an NMDA spike, $p_{NMDAsp}$, was calculated as follows:

$$p_{NMDAsp} = \frac{n_{supra}}{60} \times 100\%$$

The amplitude of an NMDA spike, $A_{NMDAsp}$, was expressed as the percentage increase compared with the amplitude of the mean linear response, $\bar{A}_{linear}$ (peak of the first Gaussian, for example, Fig. 1c):

$$A_{NMDAsp} = \left(\frac{A_{NDMAsp}}{\bar{A}_{linear}} - 1\right) \times 100\%$$

**Two-photon $Ca^{2+}$ imaging.** CA3 pyramidal neurons were loaded with Fluo-5F (100 μM) and Alexa Fluor 495 (10 μM; Molecular Probes) through the recording pipette for at least 20 min before two-photon imaging. Neurons were imaged using a two-photon microscope (Scientifica) equipped with a Ti:sapphire laser (Tsunami, Spectra Physics) tuned to 840 nm and a 40× water-immersion objective lens (0.8 NA, Olympus). Laser power under the objective was typically between 10 and 15 mW. Fluorescence was detected using two photomultiplier tubes using 525/50 nm (green channel) and 620/60 nm (red channel) emission filters. Scanning and image acquisition were controlled by HelioScan software[55]. Time series of Fluo-5F fluorescence images were acquired at 10 Hz with 100 × 100 pixel resolution across imaging fields encompassing neuronal dendrites. The dimensions of a field of view (FOV) were 30 × 30 μm.

For the imaging experiments, the ITDP protocol consisted of 60 pairings. As it was of interest to compare responses between different FOVs, images were obtained sequentially from each FOV resulting in data sets consisting of 30 responses when 2 FOVs were selected or 20 responses when 3 FOVs were selected. Simultaneous data acquisition from more than 1 FOV would have resulted in images with insufficient resolution for detailed analysis. In this respect, it should be noted that we previously found no change in the probability of evoking an NMDA spike over the time course of the 60 pairings (Supplementary Fig. 4 in Brandalise and Gerber, 2014).

At the end of each experiment, a z-stack of the fluorescently labelled CA3 pyramidal cell was acquired. Dye which inadvertently leaked from the recording pipette was sometimes taken up by surrounding glia cells. Data were analysed with NIH ImageJ and Igor Pro (WaveMetrics) software. In ImageJ, dendritic segments were manually selected as regions of interest (ROIs). $Ca^{2+}$ signals were expressed as $\Delta F/F = (F - F_0)/F_0$ where $F$ and baseline $F_0$ represent mean fluorescence values in an ROI. A $Ca^{2+}$ transient was accepted as a signal when its amplitude was greater than two times the s.d. of the noise. The onset of the evoked $Ca^{2+}$ transients was defined by the timing of the electrical stimulation (0.1 Hz). $Ca^{2+}$

transient integrals for ROIs were calculated in MATLAB as temporal integrals over a 2 s post-stimulus time window (units of '%s'; after subtraction of the mean $\Delta F/F$ in a 1 s pre-stimulus baseline window). Collapsed z-stack overview images were used to measure the distance from dendritic branches to the recording electrode in the soma. For the analysis of the frequency of $Ca^{2+}$ transients associated with NMDA spikes, as well as their amplitudes and integrals, ROIs were selected such that they enclosed an entire dendritic segment, hence the area covered by ROIs was variable (Fig. 1 and Supplementary Figs 2–5 and 7). For detailed analysis of the spatial extent of the $Ca^{2+}$ transients, a series of uniformly sized ROIs ($\sim 1 \times 1$ μm) were positioned along dendrites exhibiting responses as identified by heat maps (Fig. 1h,i and Supplementary Fig. 2h–j).

**Statistical analysis.** All data are expressed as the mean ± s.e.m. No data sets were excluded from analysis. Statistical analyses were performed using Origin 2016 (OriginLab) applying Student's paired or unpaired $t$-test. Before applying the Student $t$-test, a QQ plot was generated and the Shapiro-Wilk test was performed for each pool of data to confirm a normal distribution. Linear correlations were evaluated with the Pearson coefficient. The size of the data sets was determined based on past experience with experiments involving *in vitro* recordings in brain slices. For the main experiments $n = 7$–15, and for control experiments $n = 5$–10.

**Data availability.** The authors declare that the experimental results supporting the findings are included in the article and the Supplementary Information and are available upon reasonable request.

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

## Acknowledgements

We appreciate the technical assistance of D. Göckeritz-Dujmovic, S. Giger, H. Kasper, F. David and P. Morciano. We thank B. Gähwiler, M. Santello and F. Loup for valuable discussions and comments on the manuscript and S. Soldado Magraner for writing MATLAB routines. This work was supported by the Swiss National Science Foundation (grant: 31003A_162558) to U.G. and the NIH (grant: U01NS0905583) to J.L.

## Author contributions

F.B. performed electrophysiological and calcium imaging experiments and designed experiments. S.C. performed calcium imaging experiments. F.H. analysed data and wrote the paper. J.L. designed experiments and wrote the paper. U.G. analysed data, designed experiments and wrote the paper.

## Additional information

**Competing financial interests:** The authors declare no competing financial interests.

**Publisher's note**: 

