## [Peer Review File · Nature Communications]

Reviewers' comments:

Reviewer #1 (Remarks to the Author):

This study investigates the mechanisms underlying NMDA spike-induced LTP in hippocampus CA3 pyramidal cells. This work builds on a previous study from the same lab in which was shown that repeated and paired stimulation of CA3 and mossy fiber inputs evoke supralinear dendritic NMDA spikes and that these contribute to eliciting LTP of the CA3 synapses. Here, the authors reveal that the NMDA spikes are correlating with local dendritic Ca²⁺ transients, which are distinct from general Ca²⁺ responses or bAPs. They also provide direct proof that the synaptic NMDA-spikes are causal to the LTP, even when they are elicited by bAPs.

These are very interesting findings. The study further unveils the mechanisms underlying a type of spiking-independent LTP that was known to exist but which was not investigated into great detail. The findings are also important for a general public, as the current view of LTP is still dominated by models of spike-timing dependent LTP. Nonetheless, evidence is emerging that even in conjunction with somatic spikes, NMDA-Ca²⁺ spikes are probably the driving forces behind LTP. The fact that this is input-specific suggests that this form is Hebbian in nature, similar to the spike-timing dependent LTP. Altogether, a very important study. The paper is very well written and looks very complete. Apart from a some minor issues there are, in my opinion, no major flaws.

1. The most important point I find the separation of linear and supralinear responses. The fact that the responses follow a bimodal distribution is a strong argument, but it would be helpful if the authors could show more directly that the red traces are indeed much larger than what one would expect based on linear summation of rCA3 and MF stimulation alone. Indirectly one can read this from Fig 1B, but it will be helpful if the authors were to show individual traces of rCA3 and MF stimuli, followed by paired responses.
2. Picrotoxin was present to reduce inhibition. It is probably not essential for the intracellular mechanism, but it would helpful to know if LTP is evoked when inhibition is not suppressed? And how does this influence the NMDA-spikes?
3. It is not clear which LTP experiments were performed in the presence of Fluo-5F. Is this the main difference between Suppl. Fig 3 and Fig 1? The same for QX-314 (apart from the bAP experiments, where this was obviously excluded).

4. The sum of the integrated Ca²⁺ responses apparently correlates with the amplitude of the NMDA spike. Was the integration performed in 3D? If not, how can the authors be sure that they captured events in all dendrites? Minor issue: what are the units for the integrated Ca²⁺ responses (Suppl. Figs)?

5. The threshold to which the authors refer in the first paragraph is presumably the stimulation intensity. Though the way it reads it could be confused with a depolarization-threshold above which they are set off.

5. Minor point in Suppl. Fig 4: Incrementing instead of decrementing stimulation...

Reviewer #2 (Remarks to the Author):

This manuscript is aimed at investigating the induction of associative potentiation in CA3 neurons. Using a combination of dendritic recordings and calcium imaging, the authors report the importance of locally generated NMDA spike in the induction of LTP. They show that NMDA spikes are triggered by pairing two excitatory pathways (namely, a local input with the mossy fibre input). Interestingly, the magnitude of the NMDA spike depends on membrane potential. Finally, it is shown that the NMDA spike can be evoked by a burst of action potential. The findings reported in this manuscript are original, the paper is well written and the quality of the science is very good but I have listed below a list of specific points that should be addressed.

Specific points

1. In the methods, the authors mentioned the use of both acute and organotypic slices. First, it is not clear why. Second, there is no information about which preparation was specifically used along the manuscript (except in Supplementary Figure 12).
2. The finding that the dendritic NMDA spike is critical for inducing LTP is certainly important but one would like to know whether LTD induced by negative correlation also requires the NMDA spike.

Minor points

- 1- The title is misleading. As it is the paper is only focused on LTP. Therefore I suggest changing "plasticity" by "potentiation" in the title.
- 2- The term dSpikes is not defined in the introduction.
- 3- In the introduction, the reference to the paper by Sjostrom et al., (J Neurophysiol 2006) might be added to the list of references demonstrating the lack of plasticity without spiking activity.
4. The reference to Schiller et al., J Physiol, 1997 should be added in the introduction (first evidence for NMDA spikes).

5. The results reported in Supplementary Figure 10 are important and should be presented in the main set of figures.
6. It is not mentioned whether the value of resting membrane potential provided in the methods was obtained after compensation of the liquid junction potential or not.

Reviewer #3 (Remarks to the Author):

Whether axonal spiking output is an essential trigger for long-term synaptic plasticity in the brain, or if local dendritic spikes do the trick on their own, remains a hotly contested question in neuroscience research. Brandalise et al explore this issue in acute and organotypic slices from rats using dendritic and somatic patch-clamp recordings as well as two-photon microscopy. The authors conclude that local dendritic spikes chiefly relying on NMDA receptors, so-called NMDA spikes, are necessary and sufficient for the induction of associative LTP in CA3 pyramidal cells. The authors also find that although back-propagating action potentials (bAPs) promote LTP induction, they appear to do so because they elicit NMDA spikes that are the actual trigger of plasticity, suggesting that bAPs are not the direct trigger of plasticity in the brain. The authors argue that their findings have important implications for the spike-timing-dependent plasticity (STDP) paradigm.

This study is addressing an important and timely issue in cellular neuroscience research. The manuscript text should be improved and the figures are poorly organized. The experiments are largely carried out well, but there are a number of unclear points that should be addressed. The discourse seems one-sided and at times makes for an uninspiring read because instead of highlighting the novel ideas and findings of this study in the context of the prior literature, it largely tries to shoot down the STDP paradigm, which is not necessarily mutually exclusive to the findings at this particular synapses type with this particular induction protocol. This manuscript is possibly suitable for publication in Nature Communications after some major revisions.

MAJOR POINTS:

- 1) The discourse is not particularly well developed and it seems quite one-sided. The debate about the role of the bAP versus local dendritic spikes in triggering synaptic plasticity has largely been about classical LTP in CA1 pyramidal cells and perhaps also in neocortical layer-5 pyramidal cells. One key sticking point has been about the relevance of STDP (cited papers by Lisman and Spruston). The present manuscript, however, is about CA3 pyramidal cells and a very different induction protocol: "input timing-dependent plasticity (ITDP) protocol consisting

of a stimulation of CA3 recurrent (rCA3) axons followed 10 ms later by mossy fiber (MF) stimulation" (lines 53-55). This does not make it uninteresting, but it is important that the authors point out that this is a particular cell and synapse type with a rather unusual induction protocol. The relevance of this induction protocol for STDP is therefore unclear as well as for this ongoing debate, but this does not come across clearly in the manuscript. This should be brought out in the title of the paper, because just stating "in the hippocampus" may imply to many readers that this finding is somehow definitely general and that it applies to CA1 PCs as well, which is probably not the case. The authors therefore need to explicitly state that these results may not generalize to other cell and synapse types in the STDP plasticity paradigm. To make a parallel, much like the pre vs. postsynaptic expression of LTP debate was resolved by an emerging consensus that the locus of plasticity expression depends on e.g. animal age, synapse type, and induction protocol, it is quite likely that the resolution to the present debate regarding the role of the bAP vs. the local dendritic spike may be resolved in a similar manner: it may simply depend on context. Without a more nuanced and scholarly discussion (more references, different synapse types, compare and contrast etc), the manuscript comes across as a biased and agenda-driven paper that aims at the outset to promote one particular view (that local spikes but not bAPs matter for plasticity), and that undersells these nice results. There is an opportunity here to improve both the Intro and the Discussion. For example, on lines 227-228, the authors state "Furthermore, at these hippocampal synapses the STDP hypothesis is incorrect in that LTP can occur without action potentials". It would be useful to provide references to those papers that have made this STDP hypothesis for this particular cell and synapse type. Also, it seems the authors themselves say that the bAP can trigger NMDA spikes which can then elicit plasticity (e.g. in Abstract and on line 229, "bAPs can sometimes contribute to the triggering of NMDA spikes"), so it is unclear to me how this means that the STDP hypothesis is incorrect. It rather seems to me that this means the STDP hypothesis is actually correct, no? This line of argument should be clarified and brought out better in the Discussion.

2) It is not well defined what an NMDA spike is. In the Introduction, please define what a dSpike is (line 36). Please make clear what an NMDA spike is too -- an NMDA spike is a form of dSpike, but not all dSpikes are NMDA spikes, right? Line 61, unclear criteria, "thus meeting important criteria for NMDA spikes", please state here what the criteria are for NMDA spikes as opposed to dSpikes in general, including references. Lines 84-85, same thing "Under our experimental conditions, however, the dendritic Ca²⁺ transients corresponded to NMDA spikes, on the basis of their kinetics, amplitude, and restricted spatial propagation", but what were those kinetics, amplitudes, and propagation criteria? At what point do NMDA spikes become general dSpikes according to the authors' own criteria? This is never made clear. How come AP5 only decreases Ca²⁺ transients instead of obliterating them completely if these are truly just NMDA spikes, line 88-91 "Both the NMDA spikes and the dendritic Ca²⁺ transients 89 were greatly decreased in number after NMDAR blockade with AP5 (NMDA spikes: from 90 44.2 {plus minus} 2.7% in control, n = 13, to 4.8 {plus minus} 1.7%, n = 6, P < 0.001; Ca²⁺ transients:

from $41.9 \pm 2.3\%$ $n = 13$ to $5.1 \pm 2.7\%$ $n = 6$, $P < 0.001 \dots$ ". What's the remaining $5.1 \pm 2.7\%$? That means $1-5/42 = 12\%$ of the signal is left, suggesting that these are in fact *not* just NMDA spikes, but dSpikes due to a combination of voltage-dependent channels. On this note, the statement on lines 92-93 that "the results clearly demonstrate that these supralinear events are NMDA spikes localized to individual dendritic branches" seems both unclear and overstated, I think it should rather say something like "the results *suggest* that these supralinear events are *chiefly* NMDA spikes". Indeed, the authors are on lines 189-190 admitting that these NMDA spikes are actually not just NMDA spikes: "although NMDA spikes are the critical event triggering LTP, voltage-dependent Ca^{2+} channels can contribute to the overall dendritic Ca^{2+} signal" -- this seems to me a more nuanced take on the authors' own findings. Lines 197-198 furthermore seem intended to incorrectly imply that the authors' findings, as opposed to prior findings in the literature, are unequivocal: "experimental constraints have hampered the unequivocal identification and characterization of the underlying dendritic events" -- there are still several caveats associated with the authors' findings (e.g. Major Point 5, to mention a big one). This Major Point 2 is related to the apparent bias and agenda mentioned in Major Point 1 above, as well as to Minor Point 8 below.

3) You cannot reformulate what Hebbian learning is, because then it is no longer the learning that Hebb postulated. Lines 225-227: "Accordingly, the Hebbian learning rule can be reformulated as requiring the coincidence of presynaptic input and a branch-specific dendritic spike." Hebb said the postsynaptic cell had to fire: "When an axon of cell A is near enough to excite cell B and repeatedly or persistently takes part in firing it, ..." So he said it fires, period. It may of course be the case that Hebb was simply wrong, but please do not adjust Hebb's postulate so that his postulate is correct no matter what we neuroscientists find. Just call it something else, because otherwise the concept of Hebbian plasticity is so fluid that it technically becomes an irrefutable concept, and then it is no longer science.

4) The authors are mixing results from acute slices and from organotypic slices without making clear which results are from which preparation. The developmental stages of these preps are quite different. Since plasticity rules often depend on animal age and developmental stage, it is important that the authors throughout clearly state which precise results were from acute slices and which were from slice culture. Supp Fig 12 helps in this regard, but it does not obviate the need to be clear about the source of the results throughout the manuscript. It would be useful to provide a table that enumerates precisely which experiments were done how. Please always clearly state if results were pooled, and if so, if they were statistically indistinguishable or not.

5) Ultimately, what happens in the slice experiments has poor predictive power for what happens in vivo in the actual intact brain -- the slice preparation (whether acute or organotypic) is after all quite screwed up in many ways. The authors should therefore discuss the papers by Pawlak and Kerr eLife 2013 and compare & contrast these to Sheffield & Dombeck as well as to Gambino

Holtmaat Nature 2014 and Cichon & Gan 2015, because these were all carried out in vivo. It is a tad odd that one of the authors of the present study opts not to refer to his own paper showing bAPs in vivo (Waters & Helmchen JN 2004) -- it seems hard to argue that these bAPs do nothing when the authors themselves argue for their role in plasticity (line 229, "bAPs can sometimes contribute to the triggering of NMDA spikes"). This Major Point is related to the bias mentioned in Major Point 1; why not mention all of the relevant literature unless there is a bias?

6) Unclear STDP protocol with experimental parameters of unclear biological relevance. For STDP experiments shown in Fig. 3, please report duration and magnitude of current injections, mean {plus minus} SEM. The current injections look excessively large and long to me, e.g. panel B for "STDP 1", are those 20 ms long? In my book, they should be 2-5 ms long to qualify as STDP, otherwise depolarizations will passively propagate into dendrites, especially at these seemingly excessive current injection magnitudes. Also, for 3 a/b right, I would have expected three short current injections, not one long depolarization, since subthreshold depolarization is long known to determine LTP (see Sjostrom et al Neuron 2001; Sjostrom & Hausser Neuron 2006). This is in particular a critical problem with Supp Fig 10, where a single stronger current injection elicits not only spikes at higher frequency but also more subthreshold depolarization, thus making it impossible to disentangle the contribution of those two factors. For figure 3F, right trace, the cell is actually depolarized so strongly that it is in depolarization block, a completely unrealistic cellular state that does not happen in the intact healthy brain. It would seem relevant to the authors' argument to redo these experiments with more typical STDP parameters, otherwise it is hard to argue that this has implications for the STDP paradigm.

MINOR POINTS:

1) Line 47, please change to "Using a combination of electrophysiology and two-photon Ca²⁺ imaging techniques, we identify...", otherwise it seems to imply the use of "electrophysiological imaging techniques". Also, please add the comma.

2) Line 68, unclear metric, " a coincident NMDA spike was present in the electrophysiological recording (93.8 {plus minus} 3.5%, n = 13; Fig. 1e)", please state explicitly here what 93.8% refers to.

3) Lines 99-100, unclear statement, "...which is consistent with the generation of the EPSP at one dendritic location at a fixed distance from the soma." EPSPs are presumably always generated at a fixed distance from the soma, they don't move around, so perhaps this is not what the authors mean to say.

4) (related to minor point 3 above) The vast majority of central synapse types in the brain have multiple synaptic contacts (a notable exception is PF terminals onto Purkinje cells). If CA3 PCs

are like neocortical L5 PCs (e.g. Markram et al JPhysiol 1997), then one would not actually expect NMDA spikes to be necessarily restricted to one location, but perhaps several. The authors should comment on this and provide a suitable reference, to put their findings in context.

5) Lines 111-112, unclear statement, "rather a critical number (~10) of Ca²⁺ transients, which are indicative of NMDA spikes", why is this indicative of NMDA spikes but not just dSpikes in general? Please clarify.

6) Line 119, after "experiments", please add a comma.

7) Line 132, after "In the next experiments", please add a comma. (Commas are actually missing in many more places, e.g. lines 64 and 48, I'm too lazy to point them all out.)

8) line 163, unclear statement, "if the generated response is below threshold to evoke an NMDA spike", how do you know here if it is an NMDA spike or not? Please clarify. Is there an absolute threshold criterion?

9) Line 203, "Fig 1h", the use of cross-refs to figures in the Discussion is rarely done. Perhaps this is not necessary here.

10) Lines 222-223, please provide supporting references after sentence, "Accumulating evidence suggests that a supralinear signal is necessary to provide the strong depolarization initiating NMDA receptor-dependent Hebbian plasticity", so the readers can see how they have accumulated. Also, what does "strong" mean? bAPs are presumably stronger in terms of peak amplitude, so perhaps the authors mean ability to open up NMDA receptors. This should be clarified.

11) Line 228, "LTP can occur without action potentials", can LTD can also occur without action potentials? One of the key findings of STDP is the importance of the tLTD window for ensuring temporal competition among inputs (Song & Abbott NN 2000, also a bit in Neuron 2001). Classical Hebbian learning can't do this, because there is no LTD, so neuroscientists have had to patch it up with e.g. the BCM rule or overall normalization of inputs to get the competition. Perhaps the authors can marry their NMDA-spike LTP with the bAP and tLTD? That'd actually be interesting, rather than this one-sided attempt to 'prove' that STDP is wrong.

12) Figures 1,2,3 etc, the ordering of figure panels is peculiar, jumping all over the place. Please do across and then down, or down and then across, otherwise it becomes difficult to read. Supp Fig 5 is the only multi-panel figure I find that is organised in the 'normal' way.

13) Line 407, typo or R_{series} measurement error, it is impossible for the series resistance to be

as low as 5 MOhm for pipette resistances that are 5-6 MOhm (line 395).

14) Line 391, typo, external solution did not have 21 mM MgCl₂.

15) The authors repeatedly state that 50 μM AP5 is used (once is enough), yet never seem to state if this is concentration of the racemate or of the enantiomer. 50 μM of the D/L racemate would incompletely block NMDARs at the high Glu concentrations expected during NMDA spikes because AP5 is a competitive blocker so can be displaced by Glu, which would help explain why NMDA spikes are not fully abolished (Major Point 2 and lines 88-91).

Dear Drs. Wright and Ranade,

We thank the reviewers for their thorough evaluation of our work. Their constructive comments and helpful suggestions have allowed us to markedly improve the manuscript. We have performed all of the suggested experiments and below we give detailed responses to each of their queries.

Reviewer #1

1. The most important point I find the separation of linear and supralinear responses. The fact that the responses follow a bimodal distribution is a strong argument, but it would be helpful if the authors could show more directly that the red traces are indeed much larger than what one would expect based on linear summation of rCA3 and MF stimulation alone. Indirectly one can read this from Fig 1B, but it will be helpful if the authors were to show individual traces of rCA3 and MF stimuli, followed by paired responses.

We have added the requested traces as an inset in the lower panel of Figure 1c.

2. Picrotoxin was present to reduce inhibition. It is probably not essential for the intracellular mechanism, but it would be helpful to know if LTP is evoked when inhibition is not suppressed? And how does this influence the NMDA-spikes?

We agree that it is important that the observed mechanism should also be apparent under more physiological conditions with synaptic inhibition intact. However, as inhibition can vary significantly between experiments, it is difficult to perform a quantitative analysis of NMDA spikes with inhibition intact. As we already showed that LTP was not prevented in the absence of picrotoxin (Brandalise & Gerber, 2014), we decided for space reasons to leave out these data here. We now alert the readers to the fact that picrotoxin was present in these experiments in the first paragraph of the results.

3. It is not clear which LTP experiments were performed in the presence of Fluo-5F. Is this the main difference between Suppl. Fig 3 and Fig 1? The same for QX-314 (apart from the bAP experiments, where this was obviously excluded).

Fluo-5F was always present in experiments in which cells were imaged (Fig. 1 and associated Supplementary Figures) but not in Fig. 2 and Fig. 3. QX-314 was always present, except in the bAP experiments as noted by the reviewer. This information is now included in the methods.

4. The sum of the integrated Ca²⁺ responses apparently correlates with the amplitude of the NMDA spike. Was the integration performed in 3D? If not, how can the authors be sure that they captured events in all dendrites? Minor issue: what are the units for the integrated Ca²⁺ responses (Suppl. Figs)?

We used the Ca²⁺ transient integral as a robust measure of Ca²⁺ transient magnitude. The integral represents the temporal integral of the $\Delta F/F$ signal within an ROI over a 2-s post-

stimulus time window. The correct unit is “%s” (‘percent times seconds’). Depending on the ROI size the integral represents spatial summation over a smaller or larger area. We re-analyzed the correlation between the amplitude of the electrophysiologically recorded NMDA spike and the Ca^{2+} transient integral (Suppl. Fig. 3) using the whole field-of-view as ROI ($30 \times 30 \mu\text{m}^2$). Because slice cultures are quasi two-dimensional we believe that negligible out-of-focus $\Delta F/F$ signal, if any, was missed. Although we cannot fully exclude additional Ca^{2+} signals outside the measured field-of-view, we consider their occurrence unlikely, because several regions throughout the entire dendritic tree were screened for Ca^{2+} signals initially and we then focused only on the responsive region, which consistently was found close to the stimulation electrode.

We have corrected the units for Ca^{2+} transient integrals and clarified these issues in the Methods section and Suppl. Figs. 3 and 4. For better illustration of the correlation of the integral Ca^{2+} transient response in multiple dendritic branches and the amplitude of NMDA spikes we have now chosen a different example neuron for the plots in Supplementary Fig. 3a-e.

5. The threshold to which the authors refer in the first paragraph is presumably the stimulation intensity. Though the way it reads it could be confused with a depolarization-threshold above which they are set off.

We thank the reviewer for pointing out this problem. This is now clarified in the Results section.

6. Minor point in Suppl. Fig 4: Incrementing instead of decrementing stimulation...

Done.

Reviewer #2:

Specific points:

1. In the methods, the authors mentioned the use of both acute and organotypic slices. First, it is not clear why. Second, there is no information about which preparation was specifically used along the manuscript (except in Supplementary Figure 12).

Experiments were conducted with organotypic slice cultures, because in this preparation the dendritic tree is not lesioned. Furthermore, the quasi-monolayer structure of slice cultures facilitates imaging of the dendritic tree. However, to show that NMDA spikes and their role in LTP induction are not an exclusive feature of slice cultures, these findings were replicated in acute slices (data shown in Supplementary Figure 10). This information is now added in the Methods section.

2. The finding that the dendritic NMDA spike is critical for inducing LTP is certainly important but one would like to know whether LTD induced by negative correlation also requires the NMDA spike.

We have now performed preliminary experiments which show that our LTD protocol does indeed evoke branch restricted calcium transients that are likely to correspond to NMDA spikes (see below). Note, however, that the amplitude and the kinetics of these spikes clearly differ from those associated with LTP. Because of the preliminary nature of these results and the complexity of this issue, we prefer not to include LTD data in this manuscript.

Branch-specific dendritic Ca^{2+} transients evoked by the reverse ITDP pairing protocol are associated with induction of LTD.

(a) Subthreshold pairing protocol for LTD induction in a CA3 pyramidal cell at resting potential in which activation of a MF input is followed after 35 ms by activation of a recurrent CA3 input (ITDP LTD). The pairing stimulation is repeated 60 times at 0.1 Hz. (b) LTD is induced at stimulated CA3 recurrent synapses. Data points are values averaged over 3 min. Inset: Example traces show averaged EPSPs before (black) and after (red) repetitive pairing.

(c) Fluo-5F labeled CA3 pyramidal neuron. **(d)** Pairing- induced calcium transients were analyzed in 5 ROIs for apical dendritic branches in FOV 1. Image at right shows localized Fluo-5F $\Delta F/F$ fluorescence change for one pairing trial (arrow in e). **(e)** Example calcium transients from ROIs selected in d, recorded during 5 representative consecutive pairings (green bars). Sweeps were averaged separately for traces with and without calcium transients. **(f)** Averaged traces from electrophysiological recordings in response to the ITDP pairing protocol in which no calcium transients were detected (black trace) and in which calcium transients were detected (brown trace). Note that the area beneath the brown trace is significantly greater than that below the black trace. **(g)** Comparison of the amplitude of the calcium transients evoked by the ITDP LTD protocol (brown trace) and the ITDP LTP protocol (red trace). Note that the Ca^{2+} transients induced by the ITDP LTD protocol are significantly smaller and have a slower decay (0.63 ms for the ITDP LTD-mediated Ca^{2+} transients versus 0.32 ms for the ITDP LTP-mediated Ca^{2+}).

Minor points:

1. *The title is misleading. As it is the paper is only focused on LTP. Therefore I suggest changing "plasticity" by "potentiation" in the title.*

Agreed, the title is now modified.

2. *The term dSpikes is not defined in the introduction.*

Done.

3. *In the introduction, the reference to the paper by Sjostrom et al., (J Neurophysiol 2006) might be added to the list of references demonstrating the lack of plasticity without spiking activity.*

Done.

4. *The reference to Schiller et al., J Physiol, 1997 should be added in the introduction (first evidence for NMDA spikes).*

Done.

5. *The results reported in Supplementary Figure 10 are important and should be presented in the main set of figures.*

We thank the reviewer for his positive assessment. Nevertheless, we feel that including this figure in the main text would put too much emphasis on suprathreshold responses and bAPs, whereas the focus of our work is more on subthreshold responses.

6. *It is not mentioned whether the value of resting membrane potential provided in the methods was obtained after compensation of the liquid junction potential or not.*

Yes, the junction potential was corrected for. This is now mentioned in the Methods section.

Reviewer #3

Whether axonal spiking output is an essential trigger for long-term synaptic plasticity in the brain, or if local dendritic spikes do the trick on their own, remains a hotly contested question in neuroscience research. Brandalise et al explore this issue in acute and organotypic slices from rats using dendritic and somatic patch-clamp recordings as well as two-photon microscopy. The authors conclude that local dendritic spikes chiefly relying on NMDA receptors, so-called NMDA spikes, are necessary and sufficient for the induction of associative LTP in CA3 pyramidal cells. The authors also find that although back-propagating action potentials (bAPs) promote LTP induction, they appear to do so because they elicit NMDA spikes that are the actual trigger of plasticity, suggesting that bAPs are not the direct trigger of plasticity in the brain. The authors argue that their findings have important implications for the spike-timing-dependent plasticity (STDP) paradigm.

This study is addressing an important and timely issue in cellular neuroscience research. The manuscript text should be improved and the figures are poorly organized. The experiments are largely carried out well, but there are a number of unclear points that should be addressed. The discourse seems one-sided and at times makes for an uninspiring read because instead of highlighting the novel ideas and findings of this study in the context of the prior literature, it largely tries to shoot down the STDP paradigm, which is not necessarily mutually exclusive to the findings at this particular synapses type with this particular induction protocol. This manuscript is possibly suitable for publication in Nature Communications after some major revisions.

Major points:

1. The discourse is not particularly well developed and it seems quite one-sided. The debate about the role of the bAP versus local dendritic spikes in triggering synaptic plasticity has largely been about classical LTP in CA1 pyramidal cells and perhaps also in neocortical layer-5 pyramidal cells. One key sticking point has been about the relevance of STDP (cited papers by Lisman and Spruston). The present manuscript, however, is about CA3 pyramidal cells and a very different induction protocol: "input timing-dependent plasticity (ITDP) protocol consisting of a stimulation of CA3 recurrent (rCA3) axons followed 10 ms later by mossy fiber (MF) stimulation" (lines 53-55). This does not make it uninteresting, but it is important that the authors point out that this is a particular cell and synapse type with a rather unusual induction protocol. The relevance of this induction protocol for STDP is therefore unclear as well as for this ongoing debate, but this does not come across clearly in the manuscript. This should be brought out in the title of the paper, because just stating "in the hippocampus" may imply to many readers that this finding is somehow definitely general and that it applies to CA1 PCs as well, which is probably not the case. The authors therefore need to explicitly state that these results may not generalize to other cell and synapse types in the STDP plasticity paradigm. To make a parallel, much like the pre vs. postsynaptic expression of LTP debate was resolved by an emerging consensus that the locus of plasticity expression depends on e.g. animal age, synapse type, and induction protocol, it is quite likely that the resolution to the present debate regarding the role of the bAP vs. the local dendritic spike may be resolved in a similar manner: it may simply depend on context. Without a more nuanced and scholarly discussion (more references, different synapse types, compare and contrast etc), the manuscript comes across as a biased and agenda-driven paper that aims at the outset to

promote one particular view (that local spikes but not bAPs matter for plasticity), and that undersells these nice results. There is an opportunity here to improve both the Intro and the Discussion. For example, on lines 227-228, the authors state "Furthermore, at these hippocampal synapses the STDP hypothesis is incorrect in that LTP can occur without action potentials". It would be useful to provide references to those papers that have made this STDP hypothesis for this particular cell and synapse type. Also, it seems the authors themselves say that the bAP can trigger NMDA spikes which can then elicit plasticity (e.g. in Abstract and on line 229, "bAPs can sometimes contribute to the triggering of NMDA spikes"), so it is unclear to me how this means that the STDP hypothesis is incorrect. It rather seems to me that this means the STDP hypothesis is actually correct, no? This line of argument should be clarified and brought out better in the Discussion.

Being clear on the definition of STDP plasticity is important and thank you for catching this important weakness. Textbook definitions (e.g. Purves) state that single backpropagating sodium action potentials are necessary and sufficient for LTP induction and act by providing the depolarization necessary for the opening of NMDA channels. In our experiments in CA3 pyramidal cells, the backpropagating action potential is neither necessary nor sufficient for inducing LTP. We did not mean to imply that other factors, including bAPs, are irrelevant. In fact, we show that bAPs can facilitate LTP induction. However, our data, both with subthreshold responses as well as with a standard STPD protocol, show that the final effector for LTP induction at this synapse is the dendritic spike and not the bAP.

Are our findings peculiar to the ITDP protocol we employed? Again, we think not, as the data from experiments using a classical STDP protocol in Figure 3 are also consistent with the central role for NMDA spikes in LTP induction, even when bAPs are evoked.

Are our findings specific for this synapse and is it unlikely that they generalize to e.g. CA1 pyramidal cells? It is true that this mechanism has not been similarly demonstrated at other synapses and we have now modified the title of the paper as suggested. Nevertheless, the correlation of dendritic spikes corresponding to NMDA spikes and synaptic plasticity in CA1 pyramidal cells (Sheffield & Dombeck, 2015), in barrel cortex (Gambino et al., 2014), and in motor cortex (Cichon & Gan, 2015), suggest the mechanism we describe will not be unique to CA3 pyramidal cells.

We have now rewritten parts of the Summary and the Discussion to provide a more balanced view and have addressed each of the reviewer's criticisms.

2. It is not well defined what an NMDA spike is. In the Introduction, please define what a dSpike is (line 36). Please make clear what an NMDA spike is too -- an NMDA spike is a form of dSpike, but not all dSpikes are NMDA spikes, right? Line 61, unclear criteria, "thus meeting important criteria for NMDA spikes", please state here what the criteria are for NMDA spikes as opposed to dSpikes in general, including references. Lines 84-85, same thing "Under our experimental conditions, however, the dendritic Ca²⁺ transients corresponded to NMDA spikes, on the basis of their kinetics, amplitude, and restricted spatial propagation", but what were those kinetics, amplitudes, and propagation criteria? At what point do NMDA spikes become general dSpikes according to the authors' own criteria? (see Figure 5) This is never made clear. How come AP5 only decreases Ca²⁺ transients instead of obliterating them completely if these are truly just NMDA spikes, line 88-91 "Both the NMDA spikes and the dendritic Ca²⁺ transients 89 were greatly decreased in number after

*NMDAR blockade with AP5 (NMDA spikes: from 90 44.2 {plus minus} 2.7% in control, n = 13, to 4.8 {plus minus} 1.7%, n = 6, P < 0.001; Ca2+ transients: from 41.9 {plus minus} 2.3% n = 13 to 5.1 {plus minus} 2.7% n = 6, P < 0.001 ...". What's the remaining 5.1 {plus minus} 2.7%? That means $1-5/42 = 12\%$ of the signal is left, suggesting that these are in fact *not* just NMDA spikes, but dSpikes due to a combination of voltage-dependent channels. (AP5 competitive blocker) On this note, the statement on lines 92-93 that "the results clearly demonstrate that these supralinear events are NMDA spikes localized to individual dendritic branches" seems both unclear and overstated, I think it should rather say something like "the results *suggest* that these supralinear events are *chiefly* NMDA spikes". Indeed, the authors are on lines 189-190 admitting that these NMDA spikes are actually not just NMDA spikes: "although NMDA spikes are the critical event triggering LTP, voltage-dependent Ca2+ channels can contribute to the overall dendritic Ca2+ signal" -- this seems to me a more nuanced take on the authors' own findings. Lines 197-198 furthermore seem intended to incorrectly imply that the authors' findings, as opposed to prior findings in the literature, are unequivocal: "experimental constraints have hampered the unequivocal identification and characterization of the underlying dendritic events" -- there are still several caveats associated with the authors' findings (e.g. Major Point 5, to mention a big one). This Major Point 2 is related to the apparent bias and agenda mentioned in Major Point 1 above, as well as to Minor Point 8 below.*

We have modified the manuscript taking into account all of the above suggestions. With regard to the D-AP5 experiments, as stated in the manuscript the effects were tested in 6 cells, not 13. Furthermore, after reexamination of the data, we noticed that there was no effect in 1/6 cells in which D-AP5 was applied, probably because of a tissue perfusion problem. If this outlier were not included, the probability of observing an NMDA spike in presence of D-AP5 would be $3.1 \pm 0.8\%$ (n = 5). However, it would not be appropriate at this point (a posteriori) to change the stats from 4.8% to 3.1%. We agree with the reviewer that the incomplete block is likely to reflect the fact that D-AP5 is a competitive antagonist. Thus, in previous experiments with MK-801, NMDA spikes were completely blocked (Brandalise & Gerber, 2014; Figure 2). Therefore, as suggested, we have changed the wording in this sentence to "supralinear events are chiefly NMDA spikes".

3. You cannot reformulate what Hebbian learning is, because then it is no longer the learning that Hebb postulated. Lines 225-227: "Accordingly, the Hebbian learning rule can be reformulated as requiring the coincidence of presynaptic input and a branch-specific dendritic spike." Hebb said the postsynaptic cell had to fire: "When an axon of cell A is near enough to excite cell B and repeatedly or persistently takes part in firing it, ..." So he said it fires, period. It may of course be the case that Hebb was simply wrong, but please do not adjust Hebb's postulate so that his postulate is correct no matter what we neuroscientists find. Just call it something else, because otherwise the concept of Hebbian plasticity is so fluid that it technically becomes an irrefutable concept, and then it is no longer science.

Agreed. This sentence is now deleted.

4. The authors are mixing results from acute slices and from organotypic slices without making clear which results are from which preparation. The developmental stages of these preps are quite different. Since plasticity rules often depend on animal age and

developmental stage, it is important that the authors throughout clearly state which precise results were from acute slices and which were from slice culture. Supp Fig 12 helps in this regard, but it does not obviate the need to be clear about the source of the results throughout the manuscript. It would be useful to provide a table that enumerates precisely which experiments were done how. Please always clearly state if results were pooled, and if so, if they were statistically indistinguishable or not.

Agreed. This point was also raised by Reviewer 2, query 1. The information is now provided in the Methods section.

5. Ultimately, what happens in the slice experiments has poor predictive power for what happens in vivo in the actual intact brain -- the slice preparation (whether acute or organotypic) is after all quite screwed up in many ways. The authors should therefore discuss the papers by Pawlak and Kerr eLife 2013 and compare & contrast these to Sheffield & Dombeck as well as to Gambino Holtmaat Nature 2014 and Cichon & Gan 2015, because these were all carried out in vivo. It is a tad odd that one of the authors of the present study opts not to refer to his own paper showing bAPs in vivo (Waters & Helmchen JN 2004) -- it seems hard to argue that these bAPs do nothing when the authors themselves argue for their role in plasticity (line 229, "bAPs can sometimes contribute to the triggering of NMDA spikes"). This Major Point is related to the bias mentioned in Major Point 1; why not mention all of the relevant literature unless there is a bias?

We have now rewritten the Discussion, adding the suggested references and addressing each of these points.

6. Unclear STDP protocol with experimental parameters of unclear biological relevance. For STDP experiments shown in Fig. 3, please report duration and magnitude of current injections, mean {plus minus} SEM. The current injections look excessively large and long to me, e.g. panel B for "STDP 1", are those 20 ms long? In my book, they should be 2-5 ms long to qualify as STDP, otherwise depolarizations will passively propagate into dendrites, especially at these seemingly excessive current injection magnitudes. Also, for 3 a/b right, I would have expected three short current injections, not one long depolarization, since subthreshold depolarization is long known to determine LTP (see Sjöström et al Neuron 2001; Sjöström & Häusser Neuron 2006). This is in particular a critical problem with Supp Fig 10, where a single stronger current injection elicits not only spikes at higher frequency but also more subthreshold depolarization, thus making it impossible to disentangle the contribution of those two factors. For figure 3F, right trace, the cell is actually depolarized so strongly that it is in depolarization block, a completely unrealistic cellular state that does not happen in the intact healthy brain. It would seem relevant to the authors' argument to redo these experiments with more typical STDP parameters, otherwise it is hard to argue that this has implications for the STDP paradigm.

We thank the reviewer for pointing out this problem. We have now redone the experiments in the suggested manner and modified the corresponding figures (Fig.3 and Supplementary Fig. 9) and text accordingly.

Minor points:

1. Line 47, please change to "Using a combination of electrophysiology and two-photon Ca²⁺ imaging techniques, we identify...", otherwise it seems to imply the use of "electrophysiological imaging techniques". Also, please add the comma.

Done.

2. Line 68, unclear metric, "a coincident NMDA spike was present in the electrophysiological recording (93.8 {plus minus} 3.5%, n = 13; Fig. 1e)", please state explicitly here what 93.8% refers to.

Done.

3. Lines 99-100, unclear statement, "...which is consistent with the generation of the EPSP at one dendritic location at a fixed distance from the soma." EPSPs are presumably always generated at a fixed distance from the soma, they don't move around, so perhaps this is not what the authors mean to say.

Corrected as suggested.

4. (related to minor point 3 above) The vast majority of central synapse types in the brain have multiple synaptic contacts (a notable exception is PF terminals onto Purkinje cells). If CA3 PCs are like neocortical L5 PCs (e.g. Markram et al JPhysiol 1997), then one would not actually expect NMDA spikes to be necessarily restricted to one location, but perhaps several. The authors should comment on this and provide a suitable reference, to put their findings in context.

As suggested by the reviewer, we have added a reference (Mitra et al., 2012) indicating that CA3 pyramidal cells typically receive multiple contacts from a given neighboring cell. In fact, calcium transients associated with NMDA spikes were often observed in two or three neighboring branches (see multiple ROIs in Figure 1, Supplementary 2), but restricted to one field of view, except for three cases (Supplementary Figure 7).

5. Lines 111-112, unclear statement, "rather a critical number (~10) of Ca²⁺ transients, which are indicative of NMDA spikes", why is this indicative of NMDA spikes but not just dSpikes in general? Please clarify.

Agreed. We have now clarified this point.

6. Line 119, after "experiments", please add a comma.

Done.

7. Line 132, after "In the next experiments", please add a comma. (Commas are actually missing in many more places, e.g. lines 64 and 48, I'm too lazy to point them all out.)

Done.

8. line 163, unclear statement, "if the generated response is below threshold to evoke an NMDA spike", how do you know here if it is an NMDA spike or not? Please clarify. Is there an absolute threshold criterion?

We used plots of the area under the response, which revealed a bimodal distribution. The second peak was blocked in the presence of D-AP5, as in the ITDP protocol (Fig 1c,d). To clarify this issue we have now added these plots as insets in Figure 3 and Supplementary Figure 9.

9. Line 203, "Fig 1h", the use of cross-refs to figures in the Discussion is rarely done. Perhaps this is not necessary here.

Agreed. We have removed the cross reference.

10. Lines 222-223, please provide supporting references after sentence, "Accumulating evidence suggests that a supralinear signal is necessary to provide the strong depolarization initiating NMDA receptor-dependent Hebbian plasticity", so the readers can see how they have accumulated. Also, what does "strong" mean? bAPs are presumably stronger in terms of peak amplitude, so perhaps the authors mean ability to open up NMDA receptors. This should be clarified.

We agree that this sentence was unclear (referring to work mainly from the Spruston lab). The sentence was removed in the rewrite of the discussion.

11. Line 228, "LTP can occur without action potentials", can LTD can also occur without action potentials? One of the key findings of STDP is the importance of the tLTD window for ensuring temporal competition among inputs (Song & Abbott NN 2000, also a bit in Neuron 2001). Classical Hebbian learning can't do this, because there is no LTD, so neuroscientists have had to patch it up with e.g. the BCM rule or overall normalization of inputs to get the competition. Perhaps the authors can marry their NMDA-spike LTP with the bAP and tLTD? That'd actually be interesting, rather than this one-sided attempt to 'prove' that STDP is wrong.

This point was also raised by reviewer 2, where we provide some preliminary data. We do not yet have a clear idea of how tLTD works. The original STDP hypothesis was of great interest but had the weakness that the postsynaptic action potentials involved were not synaptically induced, but rather directly triggered by the investigator. Our data using the more physiologically realistic LTP evoked using only synaptic input indicate that the STDP rules for LTP should be modified, at least for this synapse. Providing clear evidence against a leading theory has value even if one cannot replace that theory with a complete new theory. Here, we have provided new data about the postsynaptic electrical signals necessary for LTP induction at this and possibly other synapses, which we believe is a significant contribution. A

complete new theory to replace STDP will require understanding how LTD works, but extensive further work will be necessary to achieve this goal.

12. Figures 1,2,3 etc, the ordering of figure panels is peculiar, jumping all over the place. Please do across and then down, or down and then across, otherwise it becomes difficult to read. Supp Fig 5 is the only multi-panel figure I find that is organised in the 'normal' way.

Done.

13. Line 407, typo or R_series measurement error, it is impossible for the series resistance to be as low as 5 MOhm for pipette resistances that are 5-6 MOhm (line 395).

We thank the reviewer for catching this error, which we have now corrected.

14. Line 391, typo, external solution did not have 21 mM MgCl₂.

Done.

15. The authors repeatedly state that 50 μM AP5 is used (once is enough), yet never seem to state if this is concentration of the racemate or of the enantiomer. 50 μM of the D/L racemate would incompletely block NMDARs at the high Glu concentrations expected during NMDA spikes because AP5 is a competitive blocker so can be displaced by Glu, which would help explain why NMDA spikes are not fully abolished (Major Point 2 and lines 88-91).

All experiments were done with D-AP5. This is now indicated throughout the manuscript.

Reviewers' Comments:

Reviewer #1 (Remarks to the Author):

I am happy with the all of the response of the authors to my comments/suggestions, as well as with the additions and changes to the manuscript.

Reviewer #2 (Remarks to the Author):

The paper is well improved. I am fully satisfied with the revision.

Reviewer #3 (Remarks to the Author):

The authors have carefully addressed all my points of criticism in full. This study is on an central topic in synaptic plasticity research that has been debated for years and this manuscript contributes key information to this debate that should be of interest to a broad neuroscience readership ranging from undergraduate neuroscience students to full professors. I am therefore delighted to recommend this important manuscript for publication in Nature Communications.